# GIS-Based Risk Assessment of Structure Attributes in Flood Zones of Odiongan, Romblon, Philippines

**Jerome G. Gacu** [1,2,3], **Cris Edward F. Monjardin** [1,2,4,*], **Kevin Lawrence M. de Jesus** [1,4] **and Delia B. Senoro** [1,2,4]

1   School of Graduate Studies, Mapua University, Manila 1002, Philippines
2   School of Civil, Environmental, and Geological Engineering, Mapua University, Manila 1002, Philippines
3   Civil Engineering Department, College of Engineering and Technology, Romblon State University, Liwanag, Odiongan, Romblon 5505, Philippines
4   Resiliency and Sustainable Development Center, Yuchengco Innovation Center, Mapua University, Manila 1002, Philippines
*   Correspondence: cefmonjardin@mapua.edu.ph

**Abstract:** Flood triggered by heavy rains and typhoons leads to extensive damage to land and structures putting rural communities in crucial condition. Most of the studies on risk assessment focus on environmental factors, and building attributes have not been given attention. The five most expensive typhoon events in the Philippines were recorded in 2008–2013, causing USD 138 million in damage costs. This indicates the lack of tool/s that would aid in the creation of appropriate mitigation measure/s and/or program/s in the country to reduce damage caused by typhoons and flooding. Hence, this study highlights a structure vulnerability assessment approach employing the combination of analytical hierarchy process, physical structure attributes, and existing flood hazard maps by the local government unit. The available flood hazard maps were layered into base maps, and building attributes were digitized using a geographic information system. The result is an essential local scale risk map indicating the building risk index correlated to the structural information of each exposed structure. It was recorded that of 3094 structures in the community, 370 or 10.25% were found to be at moderate risk, 3094 (76.79%) were found to be high risk, and 503 (12.94%) were very high risk. The local government unit can utilize the resulting maps and information to determine flood risk priority areas to plan flood mitigation management strategies and educate people to improve the structural integrity of their houses. A risk map gives people an idea of what to improve in their houses to reduce their vulnerability to natural disasters. Moreover, the result of the study provides direction for future studies in the country to reduce loss and enhance structure resiliency against flooding.

**Keywords:** AHP; building attributes; flood; risk assessment; GIS

## 1. Introduction

Only a few studies analyzed the risks of flood disasters based on buildings attributes, and risk assessments mostly focused on earthquakes [1], seismic areas [2], and debris flow [3]. Some studies have focused on tackling aftermath scenarios such as building damage loss [4], building damage assessment [5], building asset value [6], level of damage [7], and building vulnerability [8] that lack primitive information about the risk level these structures were at prior to the occurrence of the disaster. Some approaches have also been introduced in describing the risk of building systems that require sufficient and comprehensive knowledge, for example, using LiDAR data [9], vulnerability curve [8], indicator-based methodology [10], rapid visual screening [11], and a probabilistic approach [1]. Another study [12] of mapping the risk of a building to flood was conducted by combining flood frequency analysis, estimation of inundation depth, and damage to loss estimation, which involve substantial information that is seldom available.





Extreme weather conditions stem from the tropical climate change effect, and many uncontrolled anthropogenic activities have converted major causes of flooding in the human ecosystem [13,14]. Floods are caused by inadequate natural paths and drainages to control the overflowing of waters during excessive rainfall or super typhoons [15]; the capacity of these natural drainages was also affected by the rapid change of land use in the area [16] that reduces land infiltration. Flooding is a devastating natural disaster that frequently happens in the Philippines [17], causing severe damage to land, numerous buildings, and rural communities [18–20]. The flood events that happen in several Asian countries, including China [18], Taiwan [21], Vietnam [22], Indonesia [23,24], Malaysia [25], and Sri Lanka [26], cause losses to both life and property annually. Significant losses, due to flooding, of billions of dollars were also reported in European countries, including Germany [27], Serbia [28], and Greece [29]. Situated in a region with a typical climate and geophysical tempest, the Philippines inevitably suffers from different calamities and is a hotbed of disaster [30]. The country is exposed to typhoons yearly, and flooding is one of the most frequently occurring natural hazards that put lives and properties at risk for affected communities [16,31]. Table 1 presents the list of the costliest typhoon events that happened in the country.

**Table 1.** Summary of the costliest typhoon events in the Philippines.

| No. | Typhoon Name | Year | Damage Cost (in USD) | Reference |
|-----|--------------|------|----------------------|-----------|
| 1 | Bopha | 2012 | 753.6 million | Yu et al., 2013 [32] |
| 2 | Haiyan | 2013 | 714.3 million | Seriño et al., 2021 [33] |
| 3 | Parma | 2009 | 487.5 million | Nolasco-Javier et al., 2015 [34] |
| 4 | Nesat | 2011 | 267.9 million | Porio et al., 2121 [35] |
| 5 | Fengshen | 2008 | 241 million | Bagsit et al., 2014 [36] |

The global community demands action to focus on and achieve disaster risk reduction for people exposed to such occurrences. To accomplish this, society needs to understand the nature of flooding and typhoons' risks to these communities and their homes [16].

Natural hazards risk assessment involves different data on the built environment such as cities, buildings, urban spaces, walkways, roadways, etc., in terms of land use and land cover [37]. The flood risk assessment (FRA) identifies at-risk communities and supports mitigation decisions to augment investment benefits. High-resolution flood models and precise lot information are essential for flood risk analysis to estimate dependable outcomes for planning, preparedness, and decision-making applications [38]. However, the quantitative building risk assessment is still less studied, as the focus of most of the risk studies is mainly on people. Previous research studies focused on flood disasters' impacts on people, land, and agriculture. There are few studies about the vulnerability of a structure itself against disaster and the possible effects of flooding in the design of structures were not considered. Structures serve as the first line of defense against disasters so there is really a need to focus a risk assessment on structures too and this could help local government provide some regulations in issuing building permits in high-risk zones to ensure the resiliency of structures that are to be constructed. With the use of advanced technologies nowadays, such as remote sensing and geographical information system [39], it has become much easier to collect and analyze location-based data and combine them with other spatial data to provide significant results. Due to these technologies, it is feasible to obtain data on building parameters in a large community, making a risk assessment of buildings much easier and faster [40].

Exposure identification of lives and properties is vital in any risk assessment connected to a natural disaster. Many of the houses constructed in rural areas in the Philippines are made of light materials that are always vulnerable to the effects of flooding disasters [41]. The exposure of residential buildings as the smallest unit in rural and urban spaces to plu-

vial flooding might result in damages if they were not designed properly [40,42]. The building design is one of the most critical parameters in flood risk assessment [43]. Basement windows, doors, and underground garage entrances are familiar places where floodwater can accumulate in a building. Many buildings have already considered preparing and designing their structures to cope with the threat of flooding, but others are still not giving any importance to it. Therefore, evaluating the risk level a building is exposed to is very important to educate people on how to properly design their houses to be more resilient against flooding. Their houses are basically their first line of protection against this disaster. Identification of at what risk level a particular building would require information related to its structural integrity such as heights of the doors, windows, installation, materials used, and age of the structure, to name a few [44].

Comprehensive FRA would require reliable flood hazard data to identify locations very susceptible to flooding, which significantly affects the risk computation [45]. The Sendai Framework was used in the study to determine the risk level [46] structures are at by combining buildings' hazard, vulnerability, and exposure attributes. This framework is usually used in disaster risk reduction management (DRRM) assessment and provides quantifiable parameters for estimating risk levels in a community. However, this study used the Sendai Framework to focus on building exposure and is not specific to people. Identifying the risk level of a certain structure could also raise awareness among people on how to improve the resiliency of their houses against disasters. The compilation and evaluation of disaster damages under the Sendai Framework develop an understanding of the efficiency of implemented disaster risk reduction policies by the local government [47]. In addition, studies [48–50] have been conducted utilizing the framework considering different parameters from its components (hazard, vulnerability, and exposure).

The geographic information system is vital in flood risk assessment because the evaluation process requires spatial information [51]. This tool was proven to be effective in combining spatial information and gathered field data to provide better results. It saves human, physical, and financial resources in mapping flood disaster information [6]. In the Philippines, GIS is one of the leading technologies used nationwide for modeling and mapping flood hazards together with remote sensing (RS) [52]. Several studies successfully utilized GIS in assessing flood risk in building and rural housing in Canada [53] and China [54].

The multicriteria decision-making (MCDM) or multicriteria analysis (MCA) proves that it can hold distinctive assessments on identifying factors or parameters of a composite decision, organize the aspects into a hierarchical tree, and analyze the relationship of elements for the identified risk [55]. Many methodologies have been recommended for MCA, but the analytical hierarchy process (AHP) is the most popular and reliable tool for resolving flood risk assessment studies [56]. Several studies commenced in different Asian regions that utilized AHP in FRA studies, including Central Asia [57], South Asia [58–60], West Asia [61,62], East Asia [63–66], and Southeast Asia [67–70]. Local studies in some areas of the Philippines, such as Romblon [49], Quezon Province [71], and Davao Oriental [72], made use of AHP as part of their risk assessment methodology. The popularity of using AHP-based research studies caused it to be more straightforward in creating a model of indecision without compromising the subjective and objective features of the valuation procedure [73].

The preceding local study [49], illustrated and focused only on the spatial distribution of flooding in the municipality. With the result of this study, the local government unit can use the maps to determine flood risk priority areas and inform people about the current state of their buildings/houses against the effect of flooding. Furthermore, the result of the study delivers direction for upcoming research in the country to condense loss and improve building resiliency. The GIS-based flood risk assessment approach focuses on buildings and not on people by evaluating building attributes. A building flood risk map was developed to identify the level of risk the structures in the community are at. It could be used to educate people on how to make their houses more resilient to floods and could help LGU

to revise its land use plans. This study answers the gap in doing risk assessment focusing entirely on the vulnerability of structures which could provide references to reduce loss of life and property, and enhance community resiliency during flooding.

## 2. Materials and Methods

### 2.1. Study Area

The municipality of Odiongan, with coordinates of 12°24′4.88″ N, 121°59′2.17″ E is one of the progressive municipalities in Romblon. The said municipality, representing 12.11% of the entire Islands Province of Romblon, has a land area of 185.67 square kilometers. The study focused on the town proper and nearby barangay in the low-lying plains [51]. Figure 1 shows the imagery map of Odiongan with its barangay administrative boundaries.

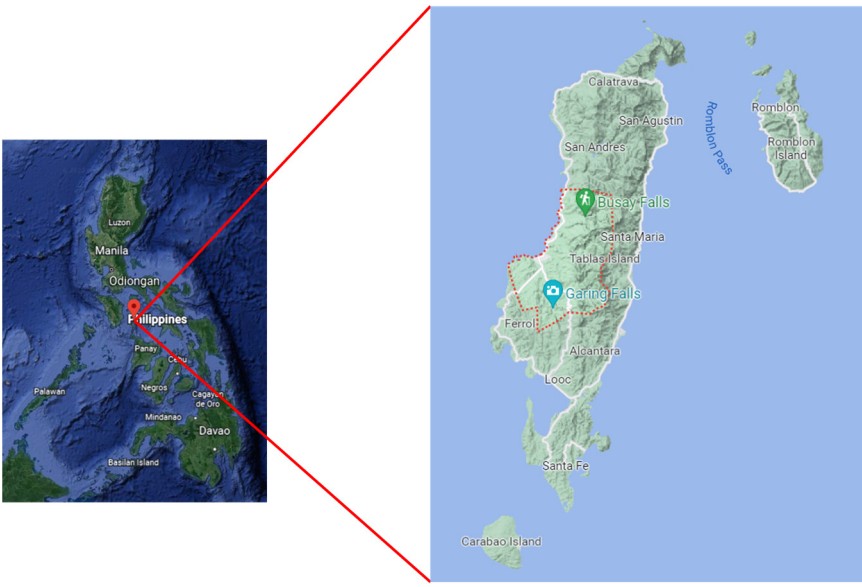

**Figure 1.** Location of the study area, Odiongan, Romblon, Philippines (12°24′4.88″ N, 121°59′2.17″ E).

According to the Corona climatic categorization system, Romblon has a climate type III, typically dry from January to May but without a clearly defined wet and dry season. The annual precipitation for the municipality of Odiongan is 1811 mm, wherein July has the highest average precipitation while February has the lowest [74]. Based on the 2022 cities and municipalities competitive index of the Department of Trade and Industry (DTI), the municipality of Odiongan ranked 81st out of 512 1st to 2nd class municipalities in terms of the infrastructure criterion, which includes components such as road network, distance to ports, availability of essential utilities, vehicles, education, health, local government unit investment, accommodation capacity, information technology capacity, and financial technology capacity. Considering the resiliency criterion, the municipality ranked 170th out of 512 1st and 2nd class municipalities in the country, which includes components such as a land use plan, disaster risk reduction plan, annual disaster drill, early warning system, budget for disaster risk reduction management, local risk assessments, emergency infrastructure, utilities, employed population, and sanitary system [75].

### 2.2. Methods of Assessing Flood Risk on Buildings

In this study, the Sendai Framework was employed to determine parameters for assessing flood risk levels focusing mainly on building exposure using the spatial and gathered field data. The conceptual framework used in this study is presented in Figure 2, showing the processes used and the relationships of identified parameters.

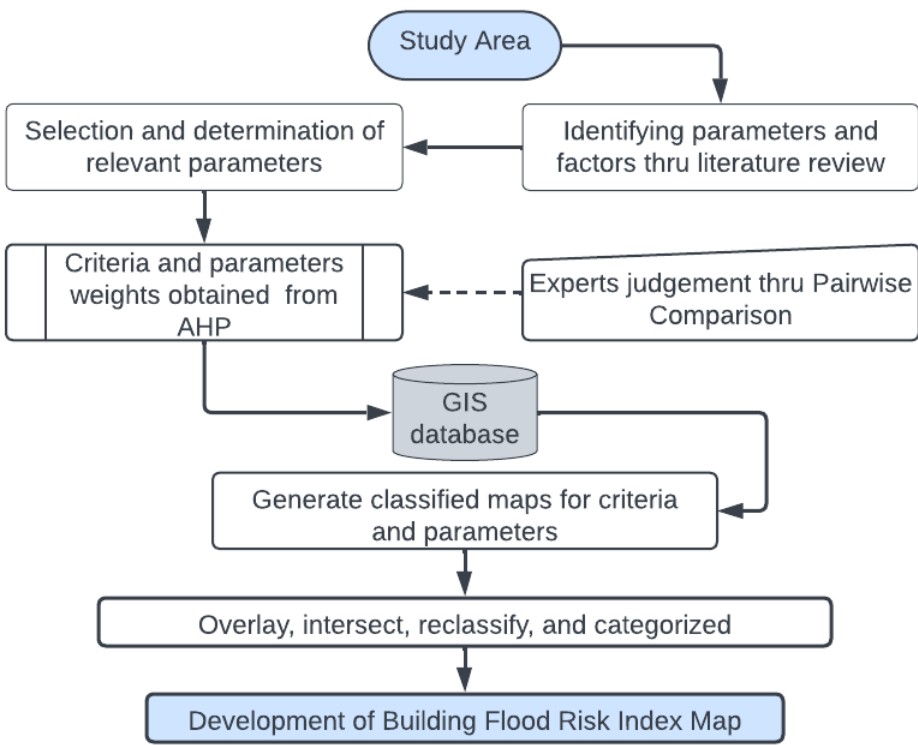

**Figure 2.** The study's methodological framework in assessing building flood risk.

Figure 2 presents the framework used in the study. Parameters in the assessment were identified through the literature review as shown in Table 2, the weights in determining hazard, vulnerability, and exposure indices were identified using an analytical hierarchy process with the help of experts consulted in this field. Data were collected through field data collection and remote sensing techniques. All these data were overlayed to produce a flood risk map focusing on the buildings that considered the three components of risk assessment, such as (a) hazard, (b) vulnerability, and (c) exposure [76].

**Table 2.** Different parameters, data types, duration/year, and sources were used in the study.

| Parameters | Data Type | Duration/ Year | Source | References |
|---|---|---|---|---|
| **Hazard Parameters** | | | | |
| Annual Average Rainfall | Interpolated Climatological Normal using Isohyetal Method | 2020 | Philippine Atmospheric, Geophysical and Astronomical Services Administration (PAGASA) | [77–79] |
| Slope | Generated from IfSAR Digital Terrian Model (DTM) using slope tool in GIS | 2013 | National Mapping and Resource Information Authority (NAMRIA) | [51] |
| Elevation | Generated from IfSAT DTM using field contour tool from GIS | 2013 | NAMRIA | [78] |
| Flood Depth | Raster file derived from existing FRA study in Odiongan, Romblon | 2022 | Previous study | [78] |

**Table 2.** *Cont.*

| Parameters | Data Type | Duration/Year | Source | References |
|---|---|---|---|---|
| **Vulnerability Parameters** | | | | |
| Average Income | Average income per barangay | 2020 | Previous study | [51] |
| Gender Ratio | Barangay men-to-women gender ratio | 2020 | Previous study | [51] |
| Land Cover | Land cover map | 2020 | NAMRIA | [80,81] |
| Roofing Material | Roofing material per building | 2022 | Field data | [82,83] |
| Flooring Material | Flooring material per building | 2022 | Field data | [82] |
| Interior Walling Material | Wall material per building | 2022 | Field data | [82] |
| Number of Floors | Number of floors per building | 2022 | Field data | [84–86] |
| Types of Fencing Material | Fencing material per building | 2022 | Field data | [86] |
| Age of Building | Age of building structure | 2022 | Field data | [84–86] |
| Total Height of Building | Earth to roof height per building | 2022 | Field data | [82] |
| Types of Windows | Window types per building | 2022 | Field data | [82] |
| Distance to River | Shapefile clipped from water courses (river) map | 2015 | United Nations Office for the Coordination of Humanitarian Affairs (OCHA) Philippines | [78,86] |
| **Exposure Parameters** | | | | |
| Building Density | Area of buildings in 50 m by 50 m land area | 2022 | Field data | [78,87] |
| Number of Buildings | Number of buildings in 50 m by 50 m land area | 2022 | Field data | [51] |
| Use of Building | List of building types according to use | 2022 | Field data | [80,88] |

The study used annual average rainfall, slope, elevation, and flood height for hazard parameters that pertain to the parameters that promote menace to buildings. Vulnerability parameters, on the other hand, consider the ability of the building and the people living in it to cope with the identified hazard. The following were used as parameters in determining the vulnerability of buildings: average income, gender ratio, land cover/use, roofing material, flooring material, walling material, number of floors, types of fencing material, age of the building, the total height of the building, types of windows, and distance to river bodies. Last is the exposure parameters, which consider the elements that might be hazardous, including population density, number of households, and buildings. All parameters with the data type and sources are shown in Table 2.

2.2.1. Building Flood Hazard Parameters

The risk assessment for buildings against flooding should be constructed based on relevant flood hazard indicators defining the impact on the surrounding areas. The parameters are as follows:

1. The study utilized the annual average rainfall map from the previous study [58], where rainfall was plotted on a base map with different stations using the isohyetal

method. The study used the climatological normal records from long-term averages over 30 years of PAGASA weather stations with corresponding coordinates.

2. The slope map was prepared using the IfSAR DTM from NAMRIA and the spatial tool in the GIS application platform [89]. IfSAR has a pixel size of 5 m × 5 m which is enough to provide accurate measurement of the slope in the area.

3. The elevation is one of the most significant factors contributing to flood hazards. Due to gravity, water flows from higher to lower elevations; therefore, low-lying areas are more prone to experience higher and longer flood duration [90]. The GIS Field Contour tool was used to process the IfSAR DTM from NAMRIA.

4. The flood susceptibility map is the most crucial parameter in hazard assessment [91]. The MGB map and modeled flood map in the study of Gacu et al. [51] were used as data for flood height. These flood susceptibility maps were based on a 100 years return period with four susceptibility levels as follows: low susceptibility to areas that could experience flood height of less than 0.5 m, moderate susceptibility for flood heights between 0.5 and 1 m, high susceptibility where flood heights are expected to reach 1 to 2 m, and last is very high susceptibility for those areas that could experience flood height of greater than 2 m [92].

### 2.2.2. Building Flood Vulnerability Parameters

Vulnerability parameters considered in this study include the evaluation of the resiliency of the building based on its design and materials used. Human factors were also considered as part of the vulnerability assessment to identify the capability to repair or improve the integrity of the building against flood disasters. Existing demographics, spatial maps, and field data were gathered through various government agencies. Field data gathering was conducted to investigate the actual state of the structures in the area.

1. Demographic data from barangay profiles, such as average income and gender ratio, were considered in the study to determine the ability of the people to provide maintenance and repair of their houses. Average income provides the capability of the people to spend on repair and maintenance while the gender ratio gives an idea of how many members living in that building could provide the manpower needed for the repairs.

2. Building attributes were surveyed manually to collect data such as building materials used, structural orientation, age, and physical dimension. The assessment focused on the building's most used material/attribute. House-to-house surveys were conducted strategically to capture data that would best represent the totality of the community. These data were digitized using GIS software to combine with other spatial data and produce a deeper assessment of building vulnerability.

### 2.2.3. Building Flood Exposure Parameters

Exposure parameters pertain to life and property features that could be exposed to flooding events. However, the focus of this study is to have a deeper exposure assessment of buildings to flood hazards. The determined exposure elements were building density, number of buildings, and type of building. Data for these exposure elements were taken from the field data.

### 2.3. Assessment of Parameters Using AHP

Identified parameters for hazard, vulnerability, and exposure components were assessed using pairwise comparison and AHP to determine the weights of each parameter based on experts' judgment [93]. Ten (10) experts in the field of DRRM participated in assessing the relevance of one parameter over the other presented in a matrix. Each parameter was graded by experts using the pairwise comparison to identify how significant it is over the other. A nine-point intensity matrix was used in the questionnaire to identify the degree of significance of one parameter over the other.

For the derivation of overall relative weights, the relative significance was calculated with the normalized values for each criterion and parameter. Normalized values for each criterion and indicator in their respective matrices originated by dividing each cell into its column, resulting in a total column of one (1) for each criterion and indicator. Weights were computed by getting the mean of the rows of the matrix. The final relative weights of the indicators were described by computing the product's linear combination (L.C.) between the relative weight of each criterion and the indicator for the specific criterion. The decision makers choose the best according to the indicators' overall weights if the experts' knowledge is recognized as consistent. Equation (1) shows the mathematical expression for the number of combinations.

$$C = \{C_j | j = 1, 2, \ldots, n\} \tag{1}$$

The pairwise comparison on *n* criteria can be simplified using the matrix (*A*) in which every element is the quotient of weights of the criteria given in Equation (2).

$$A = \begin{bmatrix} a_{11} & a_{12} & \cdots & a_{1n} \\ a_{21} & a_{22} & \cdots & a_{2n} \\ \vdots & \vdots & \cdots & \vdots \\ a_{n1} & a_{n2} & \cdots & a_{nn} \end{bmatrix}, \; a_{ii} = 1, \; a_{ji} = \frac{1}{a_{ji}}, \; a_{ij} \neq 0 \tag{2}$$

For the last mathematical process, relative weights per matrix were identified and normalized. The right eigenvector imparts the relative weights (*w*) following the highest eigenvalue ($\lambda_{\max}$) as in Equation (3).

$$A_w = \lambda_{\max} \tag{3}$$

If the pairwise comparisons were consistent, the matrix *A* has rank one and $\lambda_{\max} = n$; the weights can be generalized by normalizing any of the rows or columns of *A*. The relativeness between the items determines the consistency, and the consistency ratio (*CI*) is assumed by Equation (4).

$$CI = \frac{(\lambda_{\max} - n)}{(n-1)} \tag{4}$$

The final consistency ratio (*CR*), which permits the decision maker to accomplish whether the assessments are passably accurate, is computed as the *CI* divided by the random index (*RI*) quotient, as shown in Equation (5).

$$CR = \frac{CI}{RI} \tag{5}$$

In the last calculation, this step tells if the proportion exceeds 0.1; the judgment is considered inconsistent. So, a consistency ratio must be below 0.1 or 10%. The process is repeated if the judgment is unpredictable until the *CR* is within the wanted percentage, then the decision maker develops a conclusion concerning the assessment results.

### 2.4. Development of Building Flood Risk Map

AHP and building flood risk assessment results were put into maps for better presentation and appreciation. According to Rincón et al., maps' flood risk helps people appreciate its value [82,94]. In determining the priorities among the decision elements, feature weights were assigned to each parameter. Levels were reclassified and normalized into one (1) for the least important and five (5) for the most important. After the identification of weights, data of each parameter were combined and overlayed in a map using GIS. A risk level for each element was then developed after combining the weights, data, and attributes. The process proceeded to overlay the three (3) criteria maps (hazard, vulnerability, and exposure) with equal weights producing the building flood risk map. The resulting maps were validated based on existing flood assessments and historical flooding event records.

# 3. Results

## 3.1. Spatial Mapping of Risk Parameters

Parameters identified from the literature reviews and collected data were processed and presented into maps. Data for each parameter were gathered from different government agencies, previous studies, and field surveys. Spatial maps have a 1:30,000 scale for better visualization. There were 3976 buildings/structures assessed in this study that were analyzed spatially to determine the risk level of each against flooding.

### 3.1.1. Building Flood Hazard Parameters

Hazard parameters in this study were identified from the literature review and listed as the following: average annual rainfall [77–79], slope [51], elevation, and flood height [78]. Figure 3 shows the map of hazard parameters for the town proper of Odiongan, Romblon. Figure 3a presents the annual average rainfall using the isohyetal method in three (3) levels in which the rain intensifies from east to west of the study area. The majority of Odiongan municipality experiences an annual rainfall of 2230 mm where 3250 structures are affected, this most likely causing flooding problems in the area. There were also 560 structures that were inside the area experiencing annual average rainfall of 2240 mm and then 157 with 2220 mm.

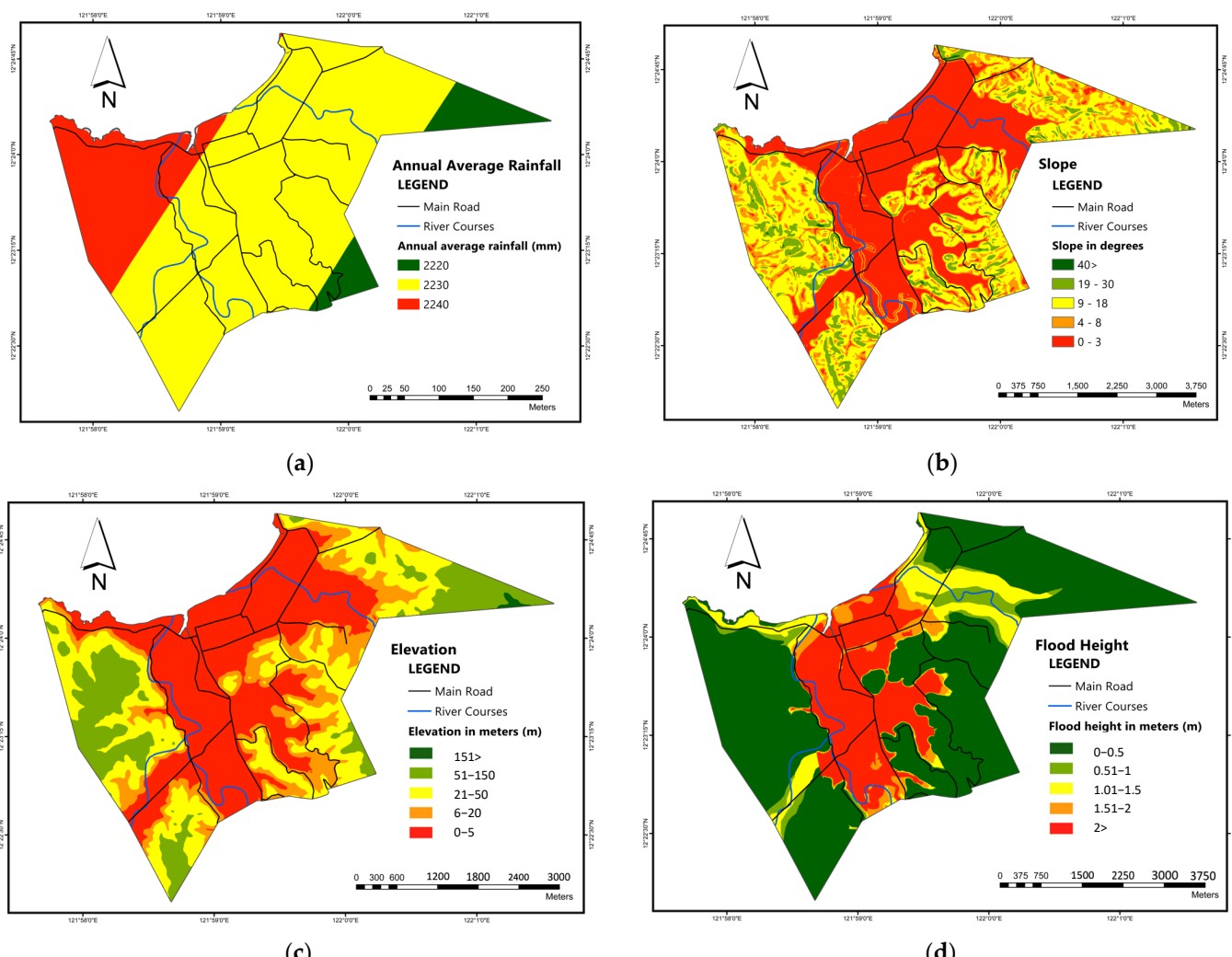

**Figure 3.** Generated maps from ArcMap in building flood hazard parameters: (**a**) average annual rainfall map, (**b**) slope map, (**c**) elevation, and (**d**) flood height map of the municipality of Odiongan.

Slope maps of the study area were categorized into five levels: <40, 19–30, 9–18, 4–8, and 0–3 with legends of green as high elevation to red as low elevation as shown in Figure 3b. The town proper itself where the majority of the residential and commercial buildings are located has a slope of 0–3 degrees covering 2762 buildings, a lower slope provides slower water velocity that could lead to water stagnation and a higher chance of flooding.

Figure 3c on the other hand shows the elevation map of the project site. Similar to slope, elevation basically presents how high the ground is with respect to mean sea level and is directly correlated to the occurrence of flooding and mean sea level is considered to be the end point of all the water flowing downstream. Area with lower elevations experiences a higher probability of flooding compared to those located in higher elevations. The elevation map was classified into five (5) categories: 0–5 m, 6–20 m, 21–50 m, 51–150 m, and 151 m and above. A total of 2855 (71.97%) buildings in the area are located in regions with a ground elevation of 0–5 m above mean sea level which puts them at risk to experience flooding.

A flood hazard map developed using hydraulic modeling combined with the susceptibility map of the Mines and Geosciences Bureau is presented in Figure 3d, five hazard levels were considered in the study, such as (1) 0–0.5 m low, (2) 0.5–1 m moderate, (3) 1.01–1.5 m high, (4) 1.51–2 m very high, and (5) 2 m and above is considered extremely high. The map shows essential information on flood height in the town proper of Odiongan. Areas near the main river experience flood heights of above 2 m which puts 1720 (43.38%) buildings near it at the most risk. There were also 1121 structures that experience flood levels from 0.5 m to 2 m. Complete records of these parameters are presented in Table A1.

3.1.2. Building Flood Vulnerability Parameters

Vulnerability analysis of buildings was based on demographics and building attributes related to being resilient of structure to flood. Figure 4 shows the spatial maps developed for the identified vulnerability parameters such as average income, gender ratio, land cover, roofing material, flooring material, walling material, number of floors, types of fencing material, age of the building, the total height of the building, types of window material, and distance to river network. Data used in these maps were gathered through actual surveys of building attributes on site and available records from local government units.

Figure 4a,b show the study area's average annual income and gender ratio per barangay zone of the municipality of Odiongan. The majority of the people in the area have an average annual income between PHP 250,000 and 499,999 that are living in 2921 (73.63%) buildings and could be seen located in low-lying areas and near the river. This was followed by residents living in 681 buildings that earn between PHP 60,000 and 99,999 located on the upstream part but still near the river while communities with a total of 365 buildings and earning less than PHP 40,000 annually are located on the left side of the town and are a little far from the river. Data showed that many economic activities happen in flat and low-lying areas since transportation is much easier which shows why many people tend to live there even if exposed to higher risk due to flooding. Average annual income was considered in the study as this gives people the capability to provide repairs and improvements to their houses for protection against flooding. The men-to-women-gender ratio was also considered as part of the vulnerability parameters since having more men is favorable as they could do heavy work and could perform repairs of their houses if damaged by disasters. Values do not differ across the municipality ranging from 0.8393 to 1.0377. There is a good ratio of men and women in the municipality; however, it can be seen at the municipal center and areas near the sea that the ratio is lower meaning more women are in those areas compared to men.

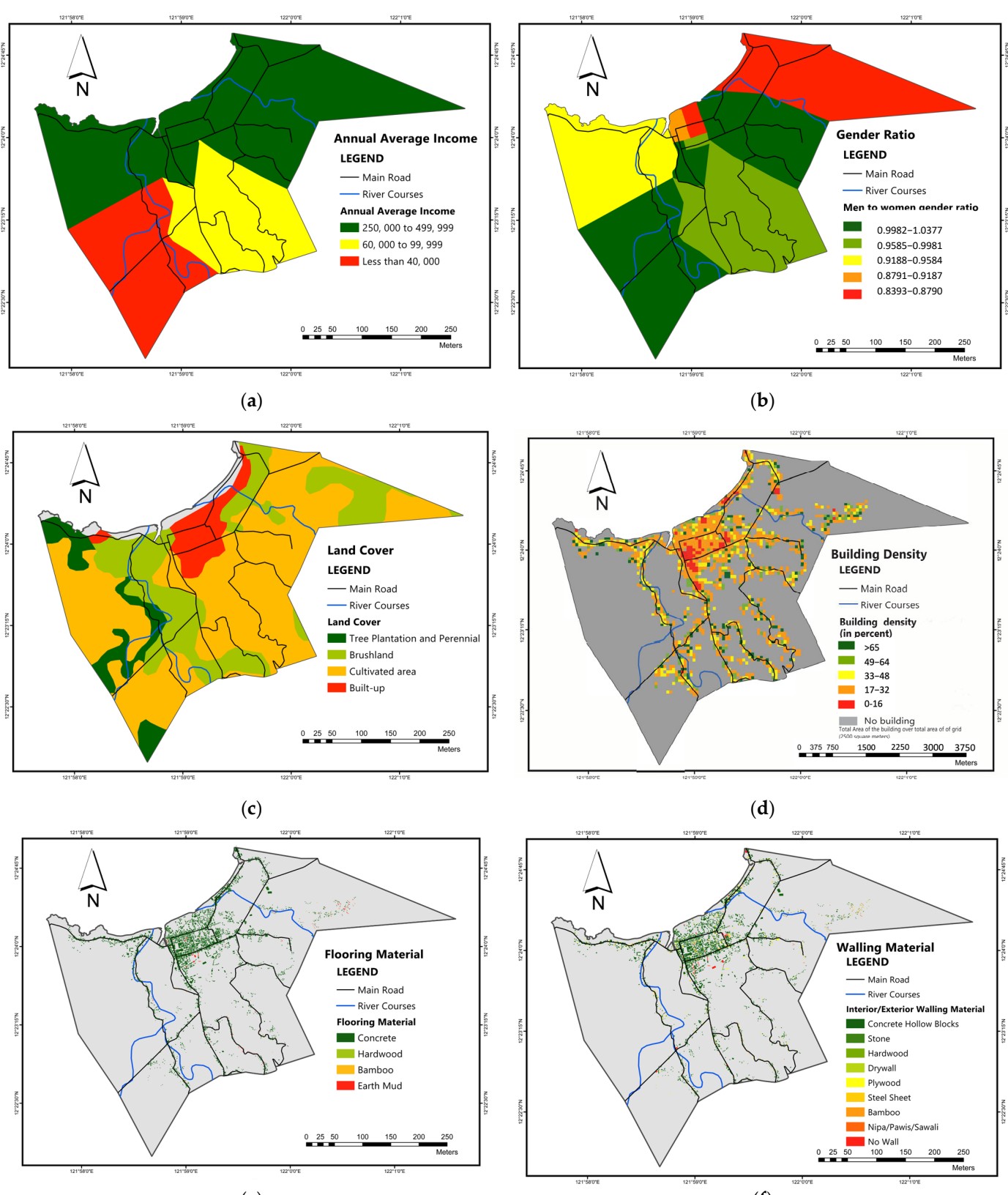

**Figure 4.** *Cont.*

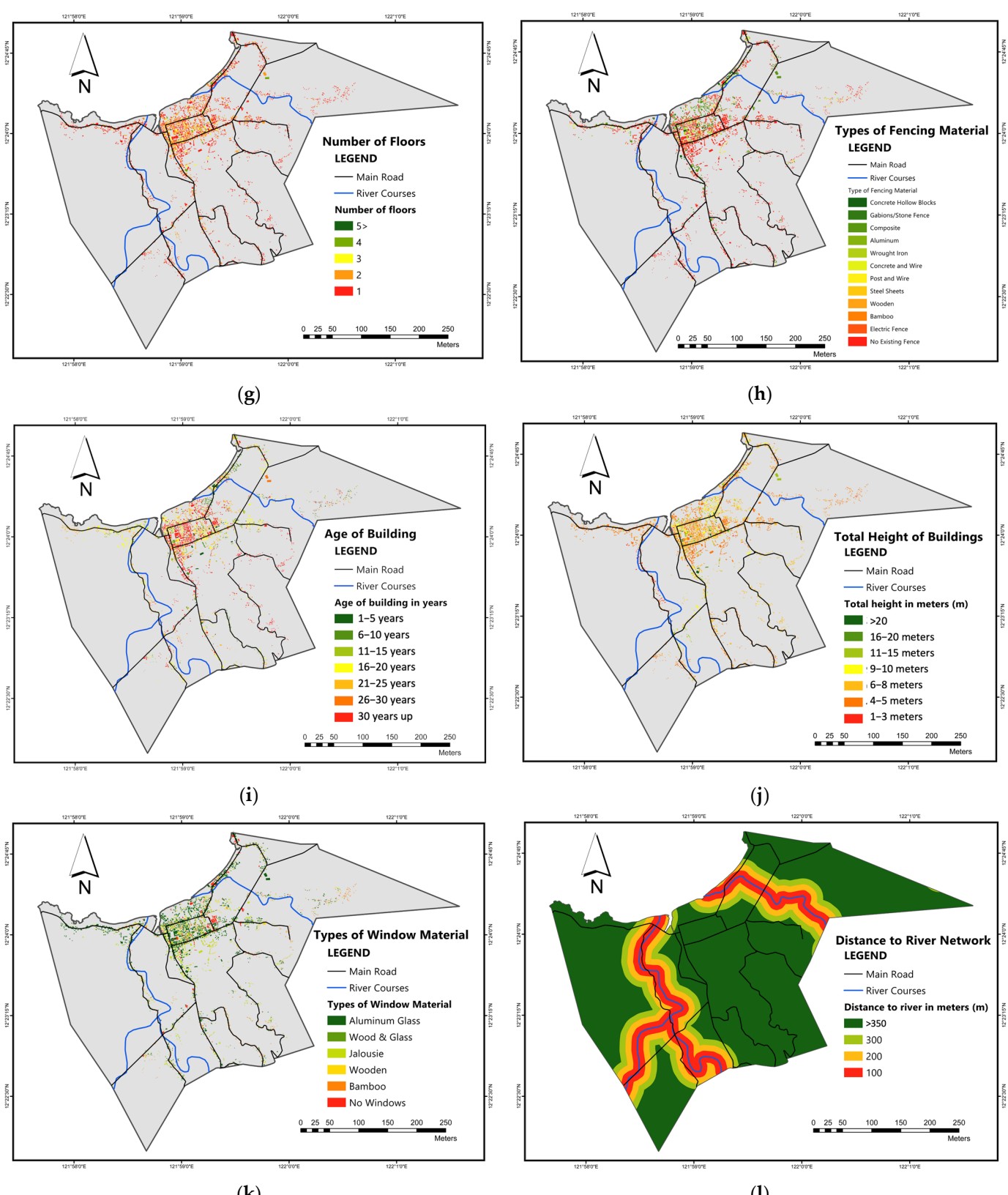

**Figure 4.** Building flood vulnerability parameters in the conduct of study: (**a**) annual average income, (**b**) gender ratio, (**c**) land cover, (**d**) types of roofing material, (**e**) types of flooring material, (**f**) interior/exterior walling material, (**g**) number of floors, (**h**) types of fencing material, (**i**) age of the building, (**j**) total height of the building, (**k**) types of window material, and (**l**) distance to river network.

Land cover was considered part of the analysis of the structure's vulnerability in this study. The location where a building was constructed is very important to identify how vulnerable it is to flooding. Developed areas are more exposed to frequent flooding due to pavements of road that reduces infiltration capability of the ground and economic activities. The study area's land cover classifications were extracted from the Comprehensive Land Use Plan (CLUP) of the municipality. The following four classifications were considered: built-up, cultivated area, brushland and tree plantation, and perennial. The map shown in Figure 4c indicates that a total of 1425 (35.92%) structures are located in built-up areas which is basically near the river and in low-lying areas, this shows poor planning that puts people and structures at risk of flooding.

Materials used in the construction of each building are an integral part of its resiliency against flooding disasters. Structures built using light materials are at higher risk to be damaged by disasters compared to those built in concrete. An actual field survey was conducted in this study to determine materials used in the construction of buildings and evaluate their resiliency to flooding. Roofing material (Figure 4d), flooring material (Figure 4e), wall material (Figure 4f), number of floors (Figure 4g), types of fencing material (Figure 4h), age of the building (Figure 4i), the total height of the building (Figure 4j), and types of window material (Figure 4k) were the parameters drawn into maps with corresponding categories and classifications. There were three thousand nine hundred sixty-seven (3967) facilities, establishments, and residential houses considered in the field data around the Poblacion (Ligaya, Liwanag, Liwayway, Tabin-Dagat, and Dapawan) and others close to town barangays such us Tulay, Bangon, and Poctoy.

Figure 4d provides information about the roofing materials of buildings in the area, 95.92% or 3805 buildings have metal sheets as their roofing material, which is a good sign that people are really investing in their houses. The flooring of these buildings was also assessed as presented in Figure 4e, a total of 3746 buildings have already used concrete as their flooring which is much stronger to resist the effect of flooding. Materials used in the wall of buildings were also evaluated, 77.84% of all the buildings have used concrete hollow blocks and the remaining 22.16% are still using light materials such as bamboo, nipa, and plywood, this is presented in Figure 4f. Figure 4g, on the other hand, presents the number of floors each building have, a building that has two or more floors is more resilient to flooding compared to those with one floor. Having multiple-level structures provides people with a place to go if the ground floor was flooded. The majority or 2763 houses in the area have only one floor which puts them at risk if a high level of flood happens. Types of fencing materials used by each house were also considered in the study as this provides protection against debris brought by flood water. A total of 2807 (70.76%) buildings do not even have a fence installed as shown in Figure 4h. The ages of buildings were also asked during the field survey and the majority or 34.51% of all the buildings were already above 30 years old. Figure 4i provides the spatial location of buildings' age which could also be a basis for local government to conduct structural integrity assessment for the safety of their community. The total height of the building as presented in Figure 4j was also measured in the study which is basically correlated to the number of floors; similarly, 49.03% (1945) of all the buildings have a height of less than 5 m. Figure 4k presents the spatial data for the type of window material used in buildings, 43.48% or 1725 buildings had aluminum and glass as their window which is resilient to the effect of flood but there was still 26.64% that used light materials such as bamboo and plywoods. The number of buildings near the river network was also determined in the study, those buildings near the river are at the most risk of flooding compared to those who are located further. There are even some who literally built their house on the riverbanks themselves. Figure 4l shows the distances of the building to the river network in four levels 100 m, 200 m, 300 m, and less than 350 m, 19.26% of all the buildings have a distance of at least 200 m from the river. These structures are the first ones to be affected by flooding disasters in the area by location. Complete records of vulnerability assessment of structures in the area could be seen in Table A2.

### 3.1.3. Building Flood Exposure Parameters

Figure 5a–c show the spatial data of the three (3) identified parameters for building exposure against flood. Building density, number of structures, and type of building use were considered as the exposure parameters. Building density was computed by getting the percent total area occupied by buildings inside a 50 × 50 m grid land area. The majority of the grids have a building density of 0–16% which is 65.32% of all the grids created in the map as shown in Figure 5a. This shows that buildings are quite far from each other and not crowding in a certain area. There were still some crowded places located inside Poblacion with a building density of greater than 65% which is 1.32% of the total grid. The Poblacion area basically is where economic activity in the municipality is centered. The number of buildings were computed by counting the number of structures in a 50 m by 50 m grid as shown in Figure 5b, 73.02% of the total grids have 1–5 structures located within. Houses and structures in provinces are naturally built far from each other unlike in the city where it is crowded everywhere. The highest number of buildings was observed in the center part of the town showing a count of 19 to 23 buildings which is in Poblacion. The use of the building was also determined during actual field data collection and used the following categories: residential, residential/commercial, institutional, agricultural, industrial, and infrastructure. Abandoned buildings were also considered for their unknown use. The majority of the buildings were residential with 2911 or 73.38% of the total. Complete records of the data gathered are presented in Table A3. Residential buildings are where people usually live and the ones that should be evaluated to ensure the safety of the people.

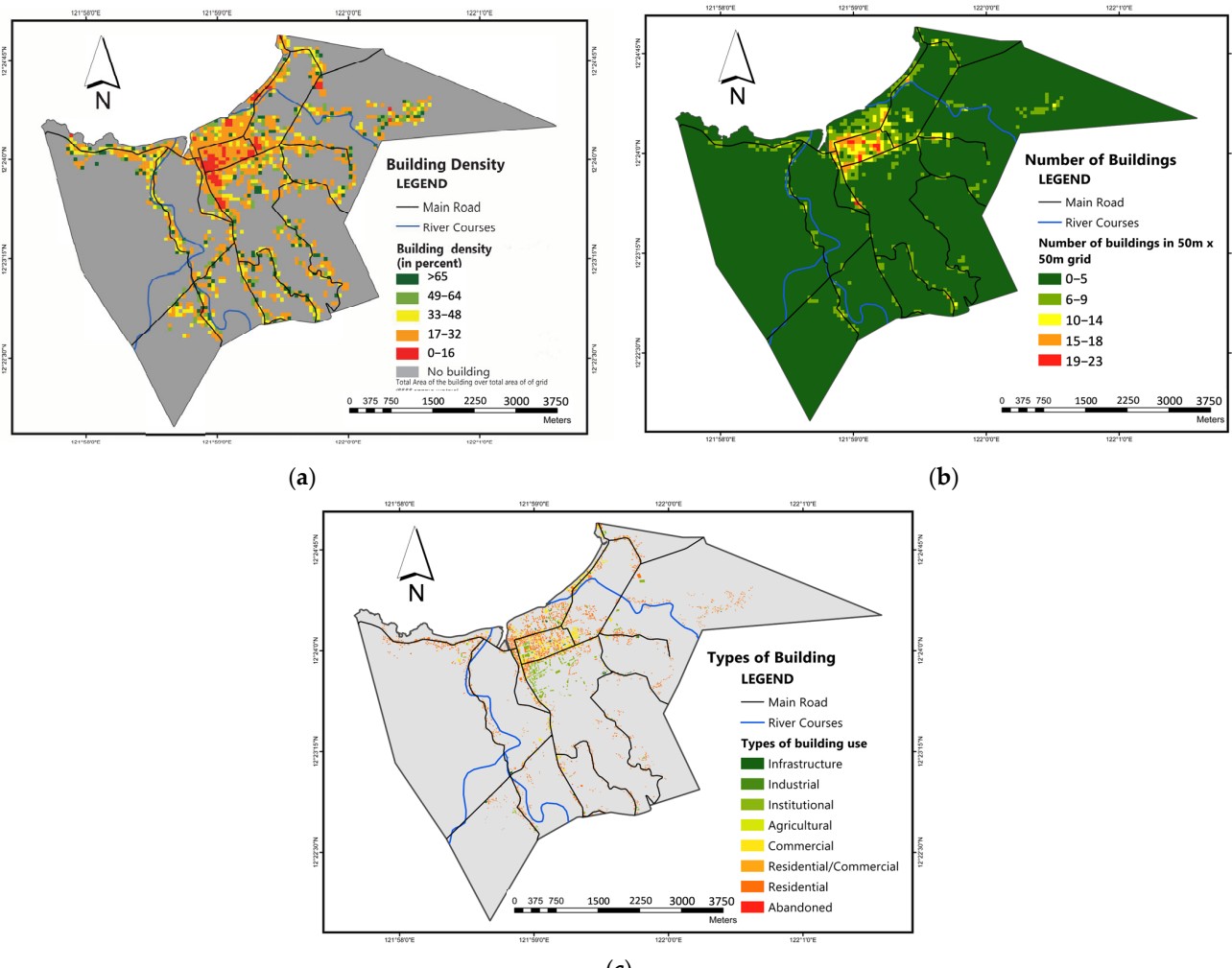

**Figure 5.** Result maps for building flood exposure parameters using ArcMap: (**a**) building density, (**b**) number of buildings per grid, and (**c**) type of building use.

### 3.2. Identification of Parameter's Weight Using AHP

Influencing parameters were evaluated and assessed using AHP and pairwise comparison techniques. The decision was separated into components and was shown in a hierarchy diagram (Figure 6) of at least three (3) levels: goal, criteria, and parameters.

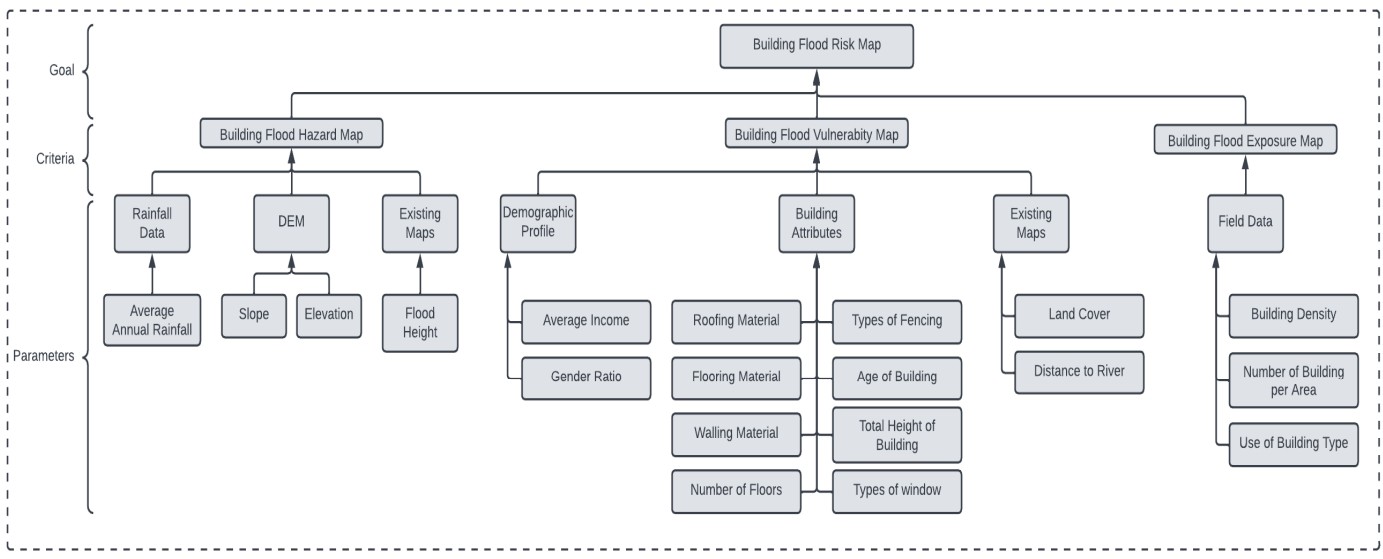

**Figure 6.** Hierarchy tree for AHP framework for building flood risk analysis.

The hierarchy tree consists of the uppermost place, the primary goal of developing a building flood risk map. The next level was the criteria for generating the goal compose of the definition of risk according to the Sendai Framework (hazard, vulnerability, and exposure), this framework was applied here specifically to building exposure to hazards. The lowest level included the parameters per criteria, substantially contributing to defining the criteria maps.

The featured class and weight were assigned for each parameter, which was reclassified and normalized using a geographic information system as shown in Table 3. Designated values depend on the type of level or category. As per results based on ten (10) experts' inputs and calculations in AHP, the final weights (percentage weights in Table 3) were identified for each and were ensured to give the CI requirement for the multicriteria analysis to be reflected valid. Flood depth had the highest weight in the building flood hazard parameters with 41.98%, almost half of the total, followed by the elevation at 30.07% then annual average rainfall at 14.6%, and lastly, slope at 13.35%. Flood depth as expected had the highest weight among flood hazard parameters since it directly provides information about the intensity of flooding in a certain area. On the other hand, twelve parameters were considered to represent the vulnerability of buildings against flooding and distance to river had the highest weight with 27.37%, followed by the total height of the building (11.91%), number of floors (11.55%), land cover (11.18%), age of the building (7.33%), roofing material (5.86%), flooring material (5.54%), wall material (4.42%), average income (4.18%), type of windows (4.02%), fencing material (3.97%), and the lowest weight of 2.67% for the gender ratio. Distance to river had the highest weight since structures near a river are expected to be exposed frequently to flooding while the height of the building and number of floors came in second and third since structures with higher height and with second floors are safer to live in. Building materials also had higher weights which showed their importance to building resiliency against flooding. Three exposure parameters were considered in the study namely building density, use of the building, and the total number of buildings with weights of 43.74%, 31.29%, and 24.97%, respectively. Building density in an area had the highest weight as it provides information on how crowded structures are in an area that could be at risk. This was followed by the type of building, residential houses are greatly affected by flooding which should be considered to be at a higher risk level compared to

other commercial and industrial establishments. The last parameter was the total number of buildings which provides information on the number of buildings that might be exposed to flood which is also an important parameter in exposure assessment.

**Table 3.** Parameters with feature class, feature weight, and percentage weight.

| Parameters | Percentage Weights (%) |
|---|---|
| **Building Flood Hazard Parameters** | |
| Annual Average Rainfall | 14.6 |
| Slope | 13.35 |
| Elevation | 30.07 |
| Flood Depth | 41.98 |
| **Building Flood Vulnerability Parameters** | |
| Average Income | 4.18 |
| Gender Ratio | 2.67 |
| Land Cover | 11.18 |
| Roofing Material | 5.86 |
| Flooring Material | 5.54 |
| Interior/Exterior Walling Material | 4.42 |
| Number of Floors | 11.55 |
| Types of Fencing Material | 3.97 |
| Age of Building | 7.33 |
| Total Height of Building | 11.91 |
| Types of Windows | 4.02 |
| Distance to River | 27.37 |
| **Building Flood Exposure Parameters** | |
| Building Density | 43.74 |
| Number of Buildings per Area | 24.97 |
| Use of Building | 31.29 |

*3.3. Development of Building Flood Risk Map*

The Sendai Framework indicates that risk is a combination of hazard, vulnerability, and exposure indices. All indices should be present to determine risk level. Figure 7a–c shows the resulting spatial map for building hazard, vulnerability, and exposure indices. All spatial maps generated from hazard, vulnerability, and exposure parameters were processed and combined together to produce the final indices of each. The weights of each parameter identified from AHP were considered in the computation of these indices. Data from all spatial maps in Figures 3–5 were combined using a 50 m grid size. Each grid extracted the data from all the spatial maps and these data with weights were used to compute the indices. There were five index levels for each: very low (green), low (yellow-green), moderate (yellow), high (orange), and very high (red) as shown in Figure 7a–c.

The resulting building flood hazard index map is presented in Figure 7a, covering 2, 424, 813, 1076, and 1652 buildings identified as very low, low, moderate, high, and very high, respectively. It was spotted that areas in the hazard index with high values were mainly affected by the flood depth map, where river networks were located. It is alarming that 68.77% of all the buildings in the municipality of Odiongan have a hazard index of high to very high.

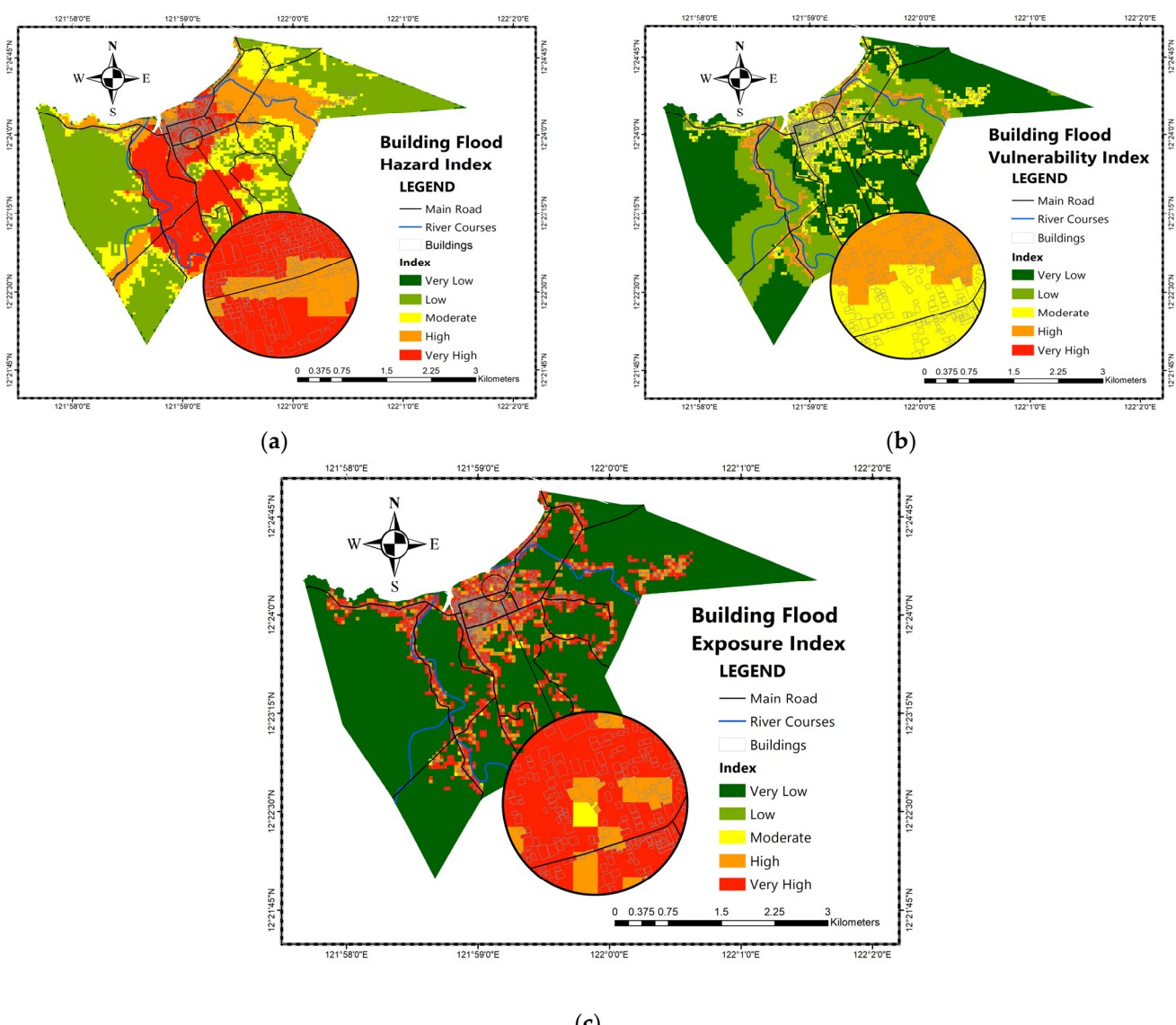

**Figure 7.** Generated maps from the different parameters in (**a**) building flood hazard, (**b**) flood vulnerability, and (**c**) flood exposure of the town proper of Odiongan, Romblon.

Figure 7b on the other hand shows the building vulnerability map. This spatial map was obtained by combining all twelve (12) parameters with five categories that connote very low vulnerability (0), low (221), moderate (2825), high (920), and very high (1). It could be seen that 71.21% of all the buildings have a vulnerability index level of moderate. This means that the majority of the buildings built in the area have moderate capability to adapt to the effect of flooding disasters. This is something that could be improved by educating the people.

Figure 7c displays the flood exposure map generated considering three (3) parameters: building density, the total number of buildings, and building use. Exposure level was presented using the following categories very low (0), low (11), moderate (1216), high (2590), and very high (15). It can be seen that 69.07% of all the buildings have a high to very high exposure index. A detailed count of buildings categorized at each level of hazard, vulnerability, and exposure index is presented in Table A4.

The generated results of building flood risk assessment were put into a map, as shown in Figure 8.

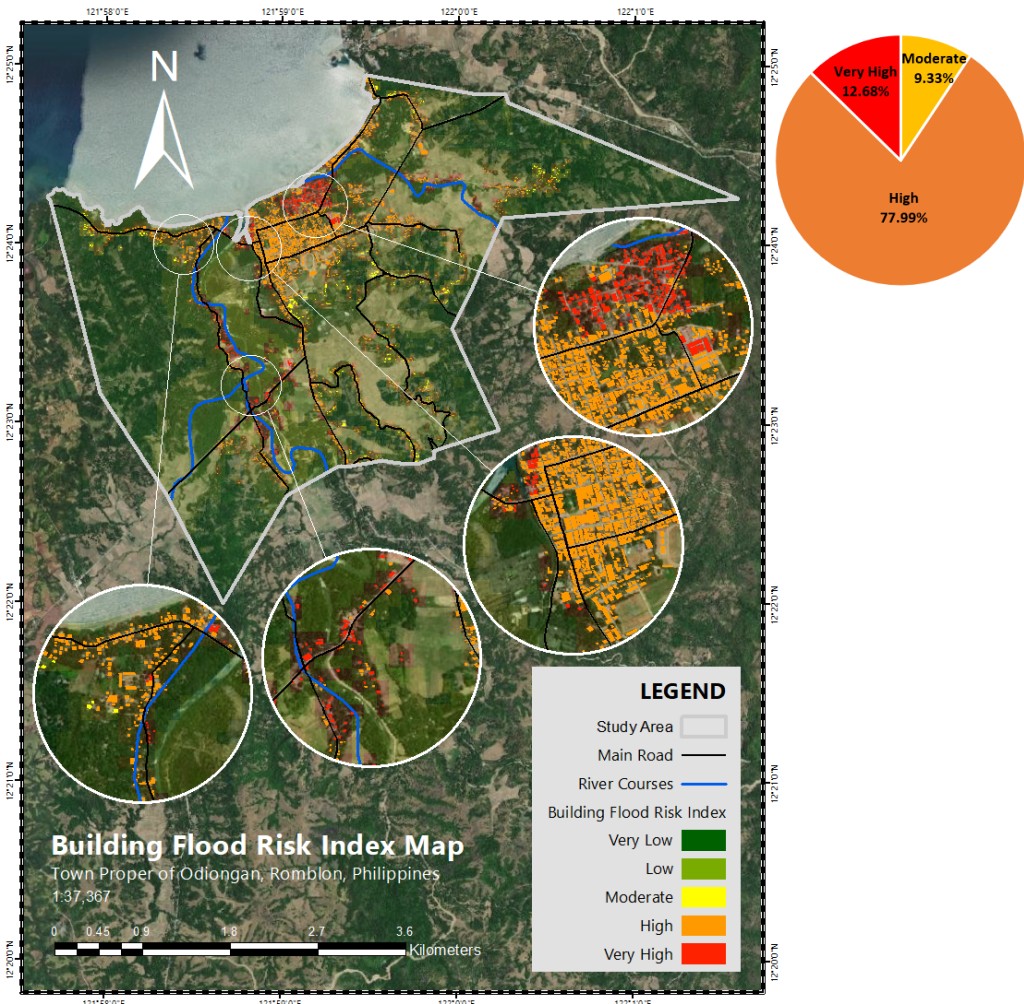

**Figure 8.** Building flood risk map with imagery and 1:15,000 scale zoomed-in map of concentrated areas with moderate to high risk of the town proper of Odiongan, Romblon.

The building flood risk map was categorized into five (5) levels (very low, low, moderate, high, and very high). The risk levels for each building/structure in the area were identified and presented in Figure 8. Out of 3967 buildings, there were 0 buildings at very low and low risk, 370 (9.33%) at moderate risk, 3094 (77.99%) at high risk, and 503 (12.68%) at very high risk. Buildings categorized to be high risk to very high risk are concentrated in the Poblacion area, specifically Ligaya, Liwanag, Liwayway, Tabin-Dagat, Budiong, and Dapawan barangay zone, these areas were located at the center of the municipality and basically located at high flood hazard zones, too. Structures in the area had high-risk levels not only because it is located in flood-prone areas but also because of the structural integrity of their buildings. The result of this study could help the community understand the risk level their structure/building/houses are at so they could do needed renovations and improvements to adapt to flooding disasters. This map could also aid the local government in updating their land use planning to avoid building structures in high-risk areas. They could also use this map to identify what type of building should be allowed to be constructed in the area based on the flood risk level.

## 4. Discussion

The study highlighted the risk assessment of floods in the building, incorporating its attributes and materials as primary building features in risk assessment. In recent years, it has become more common to integrate the AHP with GIS and remote sensing, allowing a variety of parameters such as hydrological, geographic, and socioeconomic data to be

taken into account with no limits on the number of input criteria [95]. Using the AHP to determine the parameter's weights, it was observed that the trend of building flood hazard attributes is FD > E > AAR > S. This is the trend in terms of percentage weights calculated with flood depth (41.98%) which is the most influential parameter. While slope (14.6%) is the least significant feature among the parameters of buildings. Flood depth basically indicated the possible flood height that could occur in the area considering different rainfall scenarios. In the study of Gacu et al. [51], flood depth had a high percentage in risk assessment focusing on people at 21.88%. This study showed that risk assessment focusing on people has comparable results to risk assessment focusing on building features. Building flood vulnerability parameters on the other hand have trend DR > THB > NF > LC > AB > RM > FM > IEWM > AI > TW > TFM > GR with distance to the river (27.37%) as the most significant parameter in terms of the percentage weights. While the gender ratio (2.67%) is the least important parameter in the building flood vulnerability assessment. Distance to the river evidently is the most influential hazard parameter. The recorded relationship in this study is similar to the work of Jamrussri et al. [96] in Thailand which showed that the highest evacuation was observed in zones nearest to the river. Lastly, for the building flood exposure parameters, the trend of the relative importance in terms of the percentage weights was observed to be BD > UB > NBA wherein the building density (43.74%) is the most influential parameter and the number of buildings per area (24.97%) is the least significant parameter. Building density is the ratio of the building floor area to the land area of concern. This provides information on how crowded the specific area. This result is comparable to the study of Duan et al., [97] that showed global flood risks were high in crowded areas around the world. This information will aid in policy-making related to the planning and development of a small or big government unit.

Building attributes and materials used as parameters combined with topographical and disaster-related data support the development of a building flood risk map. This map is essential in decision-making for city planning and climate change adaptation measures by classifying risk on buildings and determining the priority of risky buildings. The results of the study indicate that the use of AHP is suitable for local risk studies, and this is similar to the findings of Swain et al. in 2020 [98].

## 5. Conclusions

In this research, a building flood risk zone-based technique was developed using AHP-GIS to produce a trustworthy risk assessment of building characteristics. This study used local field data to propose a hybrid multicriteria approach. The study assessed the risk level of structures against flooding in the town proper of Odiongan, Romblon, considering building attributes and materials following the Sendai Framework in risk assessment. The study utilized GIS to map field data and perform spatial analysis. The following hazard parameters were considered such as annual average rainfall, slope, elevation, and flood depth. Building vulnerability index on the other hand includes average income, gender ratio, land cover, roofing material, flooring material, interior/exterior walling material, number of floors, types of fencing material, age of the building, the total height of the building, types of windows, and distance to the river. The exposure index covered three parameters: building density, use of a building, and the number of buildings. Each parameter was compared to one another by pairwise comparison to compute the final weights using AHP based on experts' decisions. In the hazard index, flood depth had the highest weight with 41.98% which pertains mainly to possible flood height that could occur in the area, while the building's distance to the river had the highest weight with 27.37%, which corresponds to the records of LGU that most of the residents evacuating during a disaster were those living near the rivers. Lastly, for the exposure index, building density had the highest weight of 43.74% which is a measure of how much land area are being occupied by buildings in a certain zone. Combining all these indices provides the risk level of buildings against flooding in the municipality of Odiongan. No buildings were categorized to be at very low and low risk, while 370 (10.25%) buildings were found to be at

moderate risk, 3094 (76.79%) at high risk, and 503 (12.94%) at very high risk. The majority of the buildings that have risk levels from high to very high are located in the Poblacion area where high flood level is also expected to occur and the resiliency of structures are also low. This building flood risk map is essential for the LGU's planning and development strategy, to improve their zoning plan and requirements for issuing building permits and ensure that structures are resilient enough against disaster. Educating people about their structures' risk level would be an eye-opener for them to make necessary improvements to their buildings. Commonly, risk assessment focused on people but buildings/structures are the first line of defense of the community against disaster which should also be taken into consideration. The methodology developed for evaluating building risk level is a first step in making or helping communities be more resilient against natural disasters. This study could serve as a reference to conduct building risk assessments applicable to countries with similar conditions.

**Author Contributions:** Conceptualization, C.E.F.M., J.G.G., and K.L.M.d.J.; methodology, C.E.F.M. and J.G.G.; software, C.E.F.M., J.G.G., and K.L.M.d.J.; validation, C.E.F.M. and J.G.G.; formal analysis, C.E.F.M. and K.L.M.d.J.; investigation, C.E.F.M. and J.G.G.; resources, C.E.F.M. and D.B.S.; data curation, C.E.F.M. and J.G.G.; writing—original draft preparation, J.G.G.; writing—review and editing, C.E.F.M., K.L.M.d.J., and D.B.S.; visualization, C.E.F.M. and K.L.M.d.J.; supervision, C.E.F.M.; project administration, C.E.F.M.; funding acquisition, C.E.F.M. All authors have read and agreed to the published version of the manuscript.

**Funding:** This research received no external funding.

**Data Availability Statement:** All data are contained in the manuscript.

**Acknowledgments:** This is to acknowledge the support in-kind of Mapua University and Romblon State University.

**Conflicts of Interest:** The authors declare no conflict of interest.

## Appendix A

**Table A1.** Feature count of the building flood hazard parameters.

| Parameters | Feature Class | Number of Buildings | Percentage % |
|---|---|---|---|
| **Building Flood Hazard Parameters** | | | |
| Annual Average Rainfall | 2220 mm | 157 | 3.96 |
| | 2230 mm | 3250 | 81.93 |
| | 2240 mm | 560 | 14.12 |
| Slope | 40 degrees> | 10 | 0.25 |
| | 19–30 degrees | 186 | 4.69 |
| | 9–18 degrees | 666 | 16.79 |
| | 4–8 degrees | 343 | 8.65 |
| | 1–3 degrees | 2762 | 69.62 |
| Elevation | 151 m> | 7 | 0.18 |
| | 51–150 m | 14 | 0.35 |
| | 21–50 m | 757 | 19.08 |
| | 6–20 m | 334 | 8.42 |
| | 0–5 m | 2855 | 71.97 |
| Flood Depth | 0–0.5 m | 1125 | 28.36 |
| | 0.51–1 m | 265 | 6.68 |
| | 1.01–1.5 m | 324 | 8.17 |
| | 1.51–2 m | 532 | 13.41 |
| | 2 m> | 1721 | 43.38 |

**Table A2.** Feature count of the building flood vulnerability parameters.

| Parameters | Feature Class | Number of Buildings | Percentage % |
|---|---|---|---|
| **Building Flood Vulnerability Parameters** | | | |
| Average Income | 250,000 to 499,999 | 2921 | 73.63 |
| | 60,000 to 99,999 | 681 | 17.17 |
| | Less than 40,000 | 365 | 9.20 |
| Gender Ratio | 0.9982–1.0377 | 1699 | 42.83 |
| | 0.3585–0.9981 | 823 | 20.75 |
| | 0.9188–0.9584 | 371 | 9.35 |
| | 0.8791–0.9187 | 331 | 8.34 |
| | 0.8393–0.8790 | 743 | 18.73 |
| Land Cover | Tree Plantation and Perennial | 197 | 4.97 |
| | Brushland | 1057 | 26.64 |
| | Cultivated area | 1288 | 32.47 |
| | Built-up | 1425 | 35.92 |
| Roofing Material | Concrete | 132 | 3.33 |
| | Metal Sheet | 3805 | 95.92 |
| | Nipa/Pawid | 21 | 0.53 |
| | No Roof | 9 | 0.23 |
| Flooring Material | Concrete | 3746 | 94.43 |
| | Hardwood | 9 | 0.23 |
| | Bamboo | 49 | 1.24 |
| | Earth Mud | 163 | 4.11 |
| Interior/Exterior Walling Material | Concrete Hollow Blocks | 3088 | 77.84 |
| | Stone | 3 | 0.08 |
| | Hardwood | 28 | 0.71 |
| | Drywall | 128 | 3.23 |
| | Plywood | 447 | 11.27 |
| | Steel Sheet | 93 | 2.34 |
| | Bamboo | 27 | 0.68 |
| | Nipa/Pawid/Sawali | 107 | 2.70 |
| | No Wall | 46 | 1.16 |
| Number of Floors | >5 | 6 | 0.15 |
| | 4 | 36 | 0.91 |
| | 3 | 142 | 3.58 |
| | 2 | 1020 | 25.71 |
| | 1 | 2763 | 69.65 |



**Table A2.** *Cont.*

| Parameters | Feature Class | Number of Buildings | Percentage % |
|---|---|---|---|
| **Building Flood Vulnerability Parameters** | | | |
| Types of Fencing Material | Concrete Hollow Blocks | 156 | 3.93 |
| | Gabions/Stone Fence | 1 | 0.03 |
| | Composite | 441 | 11.12 |
| | Aluminum | 4 | 0.10 |
| | Wrought Iron | 225 | 5.67 |
| | Concrete and Wire | 17 | 0.43 |
| | Post and Wire | 197 | 4.97 |
| | Steel Sheets | 6 | 0.15 |
| | Wooden | 60 | 1.51 |
| | Bamboo | 52 | 1.31 |
| | Electric Fence | 1 | 0.03 |
| | No Existing Fence | 2807 | 70.76 |
| Age of Building | 1–5 years | 222 | 5.60 |
| | 6–10 years | 303 | 7.64 |
| | 11–15 years | 503 | 12.68 |
| | 16–20 years | 815 | 20.54 |
| | 21–25 years | 712 | 17.95 |
| | 26–30 years | 43 | 1.08 |
| | 30 years up | 1369 | 34.51 |
| Total Height of Building | >20 m | 7 | 0.18 |
| | 16–20 m | 32 | 0.81 |
| | 11–15 m | 185 | 4.66 |
| | 9–10 m | 301 | 7.59 |
| | 6–8 m | 1497 | 37.74 |
| | 4–5 m | 1782 | 44.92 |
| | 1–3 m | 163 | 4.11 |
| Types of Windows | Aluminum Glass | 1725 | 43.48 |
| | Wood and Glass | 11 | 0.28 |
| | Jalousie | 1174 | 29.59 |
| | Wooden | 217 | 5.47 |
| | Bamboo | 488 | 12.30 |
| | No Windows | 352 | 8.87 |
| Distance to River | >350 m | 10,581,002.65 | 69.59 |
| | 300 m | 1,696,785.76 | 11.16 |
| | 200 m | 1,502,012.00 | 9.88 |
| | 100 m | 1,425,639.62 | 9.38 |

**Table A3.** Feature count of building flood exposure parameters.

| Parameters | Feature Class | Number of Buildings | Percentage % |
|---|---|---|---|
| **Building Flood Exposure Parameters** | | | |
| Use of Building | Infrastructure | 52 | 1.31 |
| | Industrial | 17 | 0.43 |
| | Institutional | 244 | 6.15 |
| | Agricultural | 4 | 0.10 |
| | Commercial | 722 | 18.20 |
| | Residential/Commercial | 6 | 0.15 |
| | Residential | 2911 | 73.38 |
| | Abandoned | 11 | 0.28 |
| Building Density per Grid | 0–16% | 65.32% | |
| | 17–32% | 20.52% | |
| | 33–64% | 12.84% | |
| | >65% | 1.32% | |
| Number of Buildings per Grid | 1–5 | 73.02% | |
| | 6–10 | 19.23% | |
| | 11–14 | 4.62% | |
| | 15–18 | 2.54% | |
| | 19–23 | 0.60% | |

**Table A4.** Feature count of computed building exposure, vulnerability, and hazard index level.

| Index Level | Exposure | Vulnerability | Hazard |
|---|---|---|---|
| Very Low | 2 | 0 | 0 |
| Low | 424 | 221 | 11 |
| Moderate | 813 | 2825 | 1216 |
| High | 1076 | 920 | 2590 |
| Very High | 1652 | 1 | 150 |
| Total features | | 3967 | |

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
