# Peer review of "GIS-Based Risk Assessment of Structure Attributes in Flood Zones of Odiongan, Romblon, Philippines"

_buildings, doi:10.3390/buildings13020506_

Round 1
Reviewer 1 Report
Please refer to attachment

Reviewer 2 Report
The paper is interesting, but some minor problems could be addressed in its final version:
Damage Cost (in Php) – should be presented in USD or other international currency,
In the table 1 Reference could be added in bot forms Name and date + No of citation
Line 59 - …..built environment…. – it should be explained built in which sense?
“..” line 157
Fig 2 is fuzzy.
It seems unnecessary in Figure 2 to link the study diagram with its results Flood risk for building. Especially since the source of the figure does not indicate a literature reference, and it seems that the central part of the figure should be cited.
In Table 2 References should be last column not first,
In the Table 2 in Vulnerability Parameters should not be Flooring material per square meter of the building, and other factors too?
Formula 2 have some mistakes
